# A Diet Rich in Saturated Fat and Cholesterol Aggravates the Effect of Bacterial Lipopolysaccharide on Alveolar Bone Loss in a Rabbit Model of Periodontal Disease

**DOI:** 10.3390/nu12051405

**Published:** 2020-05-14

**Authors:** Alfonso Varela-López, Pedro Bullón, César L. Ramírez-Tortosa, María D. Navarro-Hortal, María Robles-Almazán, Beatriz Bullón, Mario D. Cordero, Maurizio Battino, José L. Quiles

**Affiliations:** 1Institute of Nutrition and Food Technology “José Mataix Verdú”, Department of Physiology, Biomedical Research Center, University of Granada, Avda del Conocimiento sn, Armilla, 18016 Granada, Spain; alvarela@ugr.es (A.V.-L.); mdnavarro@ugr.es (M.D.N.-H.); 2Department of Stomalogy, Dental School, University of Sevilla, C/Avicena s.n., 41009 Sevilla, Spain; pbullon@us.es (P.B.); beatrizbullon@hotmail.com (B.B.); 3UGC de Anatomía Patológica, Hospital San Cecilio de Granada, Avda, Conocimiento s/n, 18100 Granada, Spain; cesarl.ramirez.sspa@juntadeandalucia.es; 4Pathological Anatomy Department, Jaen Hospital, 23007 Jaen, Spain; maria.robles.exts@juntadeandalucia.es; 5Cátedra de Reproducción y Genética Humana del Instituto para el Estudio de la Biología de la Reproducción Humana (INEBIR), Universidad Europea del Atlántico (UNEATLANTICO)-Fundación Universitaria Iberoamericana (FUNIBER), 39011 Santander, Spain; mdcormor@us.es; 6Newcastle Institute for Ageing and Institute for Cell and Molecular Biology, Campus for Ageing and Health, Newcastle University, Newcastle University, NE4 5PL, Newcastle upon Tyne NE4 5PL, UK; 7Department of Clinical Sicences, Università Politecnica delle Marche, 60131 Ancona, Italy; m.a.battino@univpm.it; 8Nutrition and Food Science Group, Department of Analytical and Food Chemistry, CITACA, CACTI University of Vigo, 36310 Vigo, Spain; 9International Research Center for Food Nutrition and Safety, Jiangsu University, Zhenjiang 212013, China; 10College of Food Science and Technology, Northwest University, Xi’an 710069, China

**Keywords:** atherogenic, atherosclerosis, NASH, non-alcoholic fatty liver disease, periodontal diseases, periodontitis, rabbits

## Abstract

Increasing evidence connects periodontitis with a variety of systemic diseases, including metabolic syndrome, atherosclerosis, and non-alcoholic fatty liver disease (NAFLD). The proposal of this study was to evaluate the role of diets rich in saturated fat and cholesterol in some aspects of periodontal diseases in a lipopolysaccharide (LPS)-induced model of periodontal disease in rabbits and to assess the influence of a periodontal intervention on hyperlipidemia, atherosclerosis, and NAFLD progression to non-alcoholic steatohepatitis. Male rabbits were maintained on a commercial standard diet or a diet rich in saturated fat (3% lard *w*/*w*) and cholesterol (1.3% *w*/*w*) (HFD) for 40 days. Half of the rabbits on each diet were treated 2 days per week with intragingival injections of LPS from *Porphyromonas gingivalis*. Morphometric analyses revealed that LPS induced higher alveolar bone loss (ABL) around the first premolar in animals receiving standard diets, which was exacerbated by the HFD diet. A higher score of acinar inflammation in the liver and higher blood levels of triglycerides and phospholipids were found in HFD-fed rabbits receiving LPS. These results suggest that certain dietary habits can exacerbate some aspects of periodontitis and that bad periodontal health can contribute to dyslipidemia and promote NAFLD progression, but only under certain conditions.

## 1. Introduction

Periodontal diseases, including chronic periodontitis and aggressive periodontitis, are inflammatory conditions characterized by severe destruction of the supporting structures of the teeth (i.e., collagen fibers of periodontal ligament and alveolar bone), resulting in gradual weakening of the tooth-supporting tissues, which eventually leads to tooth loss. Different pathogenic mechanisms under this set of diseases have been described, but most of them typically start with colonization or infection of the biofilm that forms at the gum line by specific bacteria, resulting in an inflammatory response characterized by swelling, redness, and bleeding of the gums, which is termed gingivitis [1]. Such an inflammatory response results from multiple proinflammatory cytokines and chemokines produced by periodontal tissue cells in response to subgingival plaque bacteria components [2]. Among others, accumulated inflammatory mediators would recruit polymorphonuclear leukocytes (PMNs) [2] producing reactive oxygen species (ROS) through NADPH oxidase [3]. In addition, proinflammatory cytokines would also lead to an increased mitochondrial production of ROS [4]. Persistent production of ROS induces oxidative stress and may trigger cell death, inflammatory responses, and perturbation of tissue homeostasis [5,6,7]. In fact, accumulating evidence indicates that oxidative stress plays a central role in periodontal tissue and alveolar bone destruction [8,9,10].

Periodontal diseases are the first cause of tooth loss, but this is preceded by other common clinical manifestations, such as deepening of periodontal pockets and loss of attachment, gradually leading to tooth loosening [1,11]. Severe periodontitis threatening tooth retention affects 10–15% of adults in the majority of populations investigated and ranged from 1% (among 20- to 29-year-old) to 39% (among individuals aged 65-year-old or above), whereas moderate periodontitis affects 40–60% of adults in all populations [12]. Therefore, periodontal diseases are one of the most prevalent chronic diseases and represent a major public health problem in many countries, also considering that poor oral health conditions have a negative impact on an individual’s quality of life [13,14]. Increasing evidence connects periodontitis with atherosclerosis and cardiovascular diseases (CVDs) [15] and with non-alcoholic fatty liver disease (NAFLD) [16]. NAFLD represents a wide spectrum of conditions, ranging from non-alcoholic fatty liver (NAFL) to non-alcoholic steatohepatitis (NASH) [17], affecting up to 30% of the adult population, particularly in developed countries [18]. In addition, NAFLD is an important risk factor for different cardiovascular disease manifestations [19]. CVDs are the main cause of death in NAFLD patients [20]. Moreover, CVDs are the leading cause of mortality [21]. Therefore, a growing interest in the relationship between oral health and systemic health has arisen. 

Elucidating whether periodontal disease and chronic disorders have a causal relationship or if they only share risk factors or pathogenic mechanisms is challenging. Consumption of low-quality diets high in foods containing saturated fat and processed meat is a shared risk factor for the development of NAFLD, metabolic disorders, and CVDs [22,23,24,25]. Likewise, different observational studies in humans suggest that saturated fat intake increases the risk of periodontal disease [26,27]. Therefore, it is possible that hyperlipidemia arising from this type of diet increases susceptibility to certain periodontal diseases along with its known role in the promotion of atherosclerosis or NASH. Furthermore, inflammation occurring during periodontitis could contribute to dyslipidemia [28], a primary risk factor for all the mentioned disorders. Lastly, inflammatory mediators, ROS, or bacterial products released during periodontal inflammation may have access to the bloodstream [29,30,31] and contribute to the recruitment of immune cells or exacerbate inflammation or oxidative stress in other organs or tissues, such as the liver or endothelium [32,33].

Periodontal disease can occur naturally in a reduced number of species, such as dogs or non-human primates, but most studies have been conducted in animal models, especially in rodents, where periodontal disease was experimentally induced [34]. The presence of certain oral microorganisms found in subgingival plaque of people suffering periodontitis is considered an indispensable factor for periodontitis onset and progression. Therefore, many animal models are based on the placement of ligatures around some tooth and/or on seeding with exogenous (human) pathogens present in dental plaque biofilm [34] or on injecting some bacterial endotoxins or components, frequently lipopolysaccharide (LPS), into gingival tissues. This LPS allows great experimental control over the pathogenic stimulus since it is directly delivered to the tissues (i.e., no “colonization variable”) in a titratable manner [35]. Rabbits have been less used than rodents in periodontal research, but they are frequently preferred for cardiovascular research due to the similarities in lipoprotein metabolism between rabbits and humans and because the sensitivity of rabbits to dietary cholesterol leads to a rapid development of severe hypercholesterolemia with prominent atherosclerosis [36]. Rabbits maintained on diets rich in cholesterol and saturated fat have also shown morphological findings in the liver consistent with characteristics that are very similar to the pathophysiology of human NASH, which is very interesting because dietary induction used accurately represents the clinical and etiological contexts of this disease [37]. This is truly relevant because of the difficulties in identifying an accurate model of human NASH in rodents [38]. Taking into account the advantages of rabbits as a model of atherosclerosis and NASH, the proposal of this study was to establish a LPS-induced model of periodontitis in male New Zealand White rabbits for evaluating the role of diets rich in saturated fat and cholesterol on some aspects of periodontal diseases and the influence of a periodontal intervention on diet-induced hyperlipidemia, NASH, and atherosclerosis.

## 2. Materials and Methods 

### 2.1. Chemicals

All the chemical products and solvents, of the highest grade available, were acquired from Sigma (St. Louis, MO, USA) and Merck (Darmstadt, Germany).

### 2.2. Experimental Design

Thirty-two male New Zealand rabbits weighing 2500 g each were kept one per cage, under a 12 h light/12 h dark cycle with free access to food and water. All the animals were fed rabbit chow for 10 days. Half of the animals were maintained on a commercial standard diet (SD), while the other half received a high-fat diet (HFD) rich in saturated fat (3% lard *w*/*w*) and cholesterol (1.3% *w*/*w*) for a total experimental period of 40 days. Half of the rabbits on each diet were treated with intragingival injections of 1 μL of dilution containing LPS from *Porphyromonas gingivalis* (InvivoGen, San Diego, CA, USA) at two sites of the lingual side of the left gingiva around the first premolar 2 days per week during the experimental period. Gingival injections were performed under intramuscularly administrated anesthesia consisting of a combination of 2% xylazine hydrochloride (15 mg/Kg) and ketamine (40 mg/Kg). At the end of the experimental period, overnight food-deprived rabbits were anesthetized with sodium pentothal (16 mg/kg) at the same time of the day to avoid any circadian fluctuation and blood was collected into heparin-coated tubes by cardiac puncture and centrifuged at 1750× *g* for 10 min. After exsanguination and death of the rabbits, a fragment of a liver lobe (the same for all the animals) and mandibles were rapidly collected. Gingival tissue around the mandibular molars and premolars and the entire aortas were rapidly dissected out. Approximately 1 cm of the aortic arch and of the thoracic and abdominal regions were selected and standardized for all the animals. The rest of the mandibles were conserved for alveolar bone loss (ABL) measurement. Gingival mucosa, aorta, and liver samples were fixed by immersion in formalin at room temperature and embedded in paraffin according to the conventional process after 24–48 h. The study was performed according to the ARRIVE Guidelines. The rats were treated following the guidelines of the Spanish Society for Laboratory Animals and the experiment was approved by the Ethics Committee of the University of Granada, Spain (permit number CEEA 2009-260).

### 2.3. Blood Biochemistry

Analyses of blood samples obtained by cardiac puncture at the end of the experimental period were performed using commercial kits (Spin React, Barcelona, Spain). 

### 2.4. Histopathological Analyses

Paraffin-embedded tissue sections of aorta and liver were stained and evaluated as previously described [39,40,41,42]. NASH was evaluated according to the Necroinflammatory Grading System for Steatohepatitis and Fibrosis scoring system proposed by Yeh and Brunt [43], as in our previous study [41]. For the analyses of the different histological features, 20 high-power fields (HPF) per sample were analyzed at a magnification of 20× or 40× in a B51 Olympus optic microscope coupled with a DP-70 Olympus camera and images taken were analyzed using the image capture and analysis software AnaliSYS Image Processing (Olympus, Hamburg, Germany), which was also used for quantitative analyses of the obtained images. The presence of the fatty streak was evaluated in aorta samples and lesions were scored on a 4-point–intensity semiquantitative scale (1, absence of damage; 2, mild damage; 3, moderate damage; and 4, severe damage). Regarding gingivae, a histological analysis was performed following Bullón et al. [44]. In particular, paraffin-embedded mucosa samples were stained with Masson’s trichrome stain for evaluating fibrosis and hematoxylin and eosin (H&E) for determining mucosa thickness, distance from the periosteum of the mandibular bone to the basal membrane, and total number of mesenchymal cells existing in the deep mucosa. Lymphocytes and PMNs that were present in a small number as well as endothelial cells of the capillaries and small vessels were discarded. Inflammatory infiltrate and macrophage or histiocyte presence were also evaluated using the same images and graded as present (1) or absent (0).

### 2.5. Alveolar Bone Morphometric Analysis

Alveolar bone morphometric analysis was performed following Dos Santos Carvalho et al. [45]. After gingival mucosa removal, the left hemimandible of each rabbit was stained with methylene blue after soft tissue removal. Images of vestibular and lingual sides of each hemimandible were taken with a camera (Canon, Tokio, Japan) with 100 mm macro and annular flash on graph paper. The distance from the top of the cementoenamel junction of the tooth to the alveolar bone crest was measured in each image using the ImageJ software at three points: vestibular, mesial, and distal.

### 2.6. Statistical Analysis

Results are expressed as mean ± standard error of the mean for eight animals. Data were checked for normality and variance homogeneity by Kolmogorov–Smirnov and Levene tests, respectively. Variables showing normal distribution were analyzed for differences between experimental treatments by an analysis of variance. In case of ABL, the maximum measurement found for each hemimandible was compared between the different treatments by an analysis of covariance using the mandibular size as a covariate or confounding variable. For multiple comparisons, the Bonferroni *post hoc* test was used. For non-normal variables, the non-parametric tests, Kruskal-Wallis, and Mann-Whitney tests were used. In all analyses, significance was determined at a P-value of 0.05. Statistical analysis was performed using SPSS 24.0 for Windows (IBM, Chicago, IL, USA).

## 3. Results

### 3.1. Blood Biochemistry

Table 1 shows data of biochemical analysis. Triglycerides (TGs), LDL-cholesterol, HDL-cholesterol, and total cholesterol levels were higher in groups receiving HFD than in groups fed with normal chow. Phospholipids and TGs were higher in HFD-fed rabbits receiving LPS injections compared to their non-treated counterparts. Among rabbits non-treated with LPS, those fed on HFD showed higher plasma levels of creatinine, uric acid, and direct bilirubin than SD-fed rabbits non-treated with LPS, but no differences in total bilirubin were found between these two groups. Lastly, HFD-fed animals receiving LPS injections showed higher activities of gamma-glutamyl transferase (GGT) and higher levels of uric acid and total and direct bilirubin compared to both groups fed on standard diets. Uric acid and total bilirubin level, as well as GGT activity, were higher in HFD-fed animals treated with LPS than those found in HFD-fed rabbits that did not receive LPS injections.

### 3.2. Histopathological Findings

Animals fed with a SD did not shown any appreciable lesion at the end of the experimental period, but HFD-fed animals showed lesions at any region of aorta and higher steatosis and NASH scores compared to groups of animals fed with standard chow. At the aortic arch and thoracic region, lesion scores were greater in HFD-fed animals non-treated with LPS compared to HFD-fed animals treated with LPS (Table 2). However, although both groups of HFD-fed rabbits showed a higher number of lobular inflammation foci than those groups receiving a SD, this value was higher in those treated with LPS when HFD-fed groups were compared between each other. Regarding ballooning degeneration of hepatocytes, no differences were found between SD-fed animals non-treated with LPS and the two groups of animals maintained on HFD. Portal tract inflammation was only found in two individuals and NASH fibrosis was almost absent. Gingival epithelium thickness (Figure 1) was higher in animals fed on a standard diet receiving LPS than in the other three groups. No statistically significant differences in the distance from gingival epithelium to periosteum (Figure 2) were found between groups, although the mean values were higher in the SD + LPS group than in the other groups. Fibrosis (Figure 3) and cellularity (Figure 4) increased in animals receiving LPS injections compared to animals non-treated with LPS. A mixed inflammatory infiltrate was occasionally detected in groups treated with LPS, but differences with animals non-treated with LPS reached statistical significance only in those animals fed with HFD. However, macrophage appearing as aggregates or accumulations of foamy histiocytes were only detected in HFD-fed groups (Figure 5).

### 3.3. Alveolar Bone Loss

ABL was higher in animals receiving LPS injections compared to those non-treated with LPS (Figure 6). Moreover, LPS-treated animals fed HFD showed higher ABL than their SD-fed counterparts (Figure 6).

## 4. Discussion

The present study focused on the establishment of a LPS-induced model of periodontitis in rabbits for evaluating the influence of periodontal health on diet-induced hyperlipidemia, NASH, and atherosclerosis progression and the role of diets rich in saturated fat and cholesterol on aspects of periodontal diseases. In rodents, repeated injections of LPS from different bacteria into the gingival tissues caused ABL [46,47] in areas near to the injection sites, as well as an obvious inflammatory reaction characterized by a significant increase in infiltrated leukocytes [46,47,48], predominantly neutrophils and macrophages [47]. However, no previous studies inducing experimental periodontitis in rabbits by injecting LPS into gingiva were found. Morphometric analysis of the mandibles of the rabbits included in this study revealed that LPS injections induced ABL around the first premolar in animals receiving a SD. In addition, some samples of the same experimental groups showed an inflammatory infiltrate featured by the presence of foamy histiocytes in aggregates that were absent in all the samples from rabbits non-treated with LPS. Despite the fact that no use of rabbit models of LPS-induced periodontitis has been reported to date, some authors have induced experimental periodontitis in this animal by placing *P. gingivalis*-soaked ligatures into gingival sulcus [48,49,50]. After a period of 6 or 14 weeks, rabbits showed a bone loss of approximately 3 to 4 mm compared to controls [48,49,50]. Likewise, histological sections showed local inflammatory cell infiltration [48,49,50]. Nevertheless, consistent with our study, all rabbits did not display increased cellular infiltration compared to the controls in one of the studies [48], which suggests that rabbits could be less sensitive than rodents. Therefore, the results of the present study confirm that LPS from *P. gingivalis* alone is enough to induce destruction of alveolar bone in rabbits and suggest that it is a reproducible experimental model of human ABL.

As mentioned, associations between periodontal disease and chronic disorders could be a consequence of the presence of certain shared risk factors. Low-quality diets and excessive energy intake seem to be determinants of metabolic syndrome, NAFLD, and CVDs [23,24]. Studies in patients with periodontitis suggest a similar association with periodontal disease pathogenesis [26,27]. Blood analysis results showed that the consumption of the experimental diet for 40 days increased plasma levels of glucose, TGs, phospholipids and LDL-, HDL- and total cholesterol in rabbits. Likewise, experimental groups fed on HFD showed atherosclerotic lesions at the aorta and NASH at the liver. Overall, these findings are consistent with those found in previous studies using similar models where diets rich in saturated fat and cholesterol have an effect in the same sense on blood lipids [40,42], aortic lesions [42,44], and the liver [40]. Thus, these results demonstrated that the diet used in the present study, which was rich in cholesterol and fat, clearly induced hyperlipidemia as well as NASH and atherosclerosis in rabbits with features characteristic of humans pathologies under our experimental conditions.

Regarding possible consequences of HFD consumption or the HFD-induced hyperlipidemia in periodontium, a previous study of rabbits showed that a similar diet rich in saturated fat and cholesterol induced some histological alteration at this tissue, including decreased gingival cellularity and a higher mean score of fibrosis with collagen fibers to become thick and diffusely distributed [40]. However, these did not evidence a clear effect on susceptibility to periodontal diseases. In contrast, the experimental diet had no additional effect on gingival cellularity or fibrosis in animals treated with LPS in the present study, but a significant number of samples showed inflammatory infiltrates compared with SD-fed rabbits non treated with LPS. Likewise, even though no differences in ABL were found between animals non-treated with LPS, results of the present study would indicate that consumption of a diet rich in saturated fat and cholesterol exacerbated ABL induced by the treatment with LPS. Similarly, a diet containing 1% cholesterol, 5% lard, and 15% yolk had no effect on ABL after 14 weeks in rabbits in absence of additional stimuli, but ABL induced by the exposition to *P. gingivalis* or *A. actinomycetemcomitans* during the last 6 weeks was exacerbated [51]. Similarly, higher ABL has been found in different studies in rodent periodontitis models when animals received different diets rich in saturated fats and/or cholesterol [52,53,54,55,56,57,58]. Interestingly, macrophage presence was evident in all groups receiving the experimental diet (HFD and HFD + LPS), a feature that did not depend on LPS administration. This phenomenon could be mediated by an increase in endothelial activation, which was previously observed in rabbits fed on a similar HFD where endothelial cells tended to be activated and become plumper and more evident [40]. In fact, atherogenic lipoproteins have also been evidenced to up-regulate endothelial activation in aortic endothelial cells of the aortic lesions, as suggested by the vascular cell adhesion molecule (VCAM-1) expression [58]. This suggests that endothelial cell dysfunction induced by atherogenic lipoproteins plays an important role in monocyte adherence to endothelium and migration not only into aorta intima, but also into gingival tissue.

Furthermore, similarly to other infectious or chronic inflammatory diseases [59], periodontal diseases might alter concentration and composition of plasma lipids and lipoproteins [60], promoting atherosclerosis and NAFLD progression. In fact, studies on different experimental periodontitis rodent models suggest that this condition led to increases in TGs [61,62] and total cholesterol and to a decrease in HDL-cholesterol [62,63]. However, Ebersole et al. [64] reported an increase in serum cholesterol, but no change in TGs in a non-human primate model of ligature-induced periodontitis. However, when these animals were maintained on a high-fat diet, a transient decline in HDL-cholesterol levels was found. Similarly, studies on humans suggest that the levels of blood lipids and lipoproteins affected by periodontal disease are not always the same. Some authors reported significant correlations between periodontal status and cholesterol levels [65,66,67], whereas others found significant associations between TG level and periodontal disease [68,69]. In the present study, plasma levels of lipids suggested that LPS administration would tend to increase all blood lipid levels when animals received a HFD, but its effect was only significant for TGs and phospholipids. Similarly, an absence of the effect of periodontal interventions on blood circulating lipids has also been reported in mice with *P. gingivalis*-soaked ligatures [70]. However, in the same study, this intervention resulted in an increase in VLDL, LDL, and total cholesterol, in addition to a decrease in HDL-cholesterol in C57BL/6.KOR-Apoe (shl) mice. All these results support the contribution of periodontal disease to dyslipidemia, but only when other more important risk factors are present, including certain diets.

In relation to atherosclerosis, Ekuni et al. demonstrated lipid deposition in the descending aorta in a rat ligature-induced periodontitis model [71]. Likewise, the exposition to *A. actinomycetemcomitans* and *P. gingivalis* led to the development of atherosclerotic lesions in ApoE−/− mice [72], a well-accepted model for atherosclerosis [73]. Interestingly, lipid accumulation was found in aortas of rabbits in response to ligature placement and application of *P. gingivalis,* but no effect on serum cholesterol was found [49]. These findings suggest that the connection between oral disease and atherosclerosis would not only depend on dyslipidemia associated with periodontal diseases. Surprisingly, it seems that LPS administration diminished the HFD-induced progression of aortic lesions at the abdominal region and, importantly, in the aortic arch to some extent in our model. In this sense, the large-sized TG-rich lipoproteins (>75 nm diameter) accumulating in the plasma of alloxan-induced type I diabetic rabbits failed to penetrate into the arterial wall [74], which explains the inhibited cholesterol-induced atherosclerosis found in these rabbits [75]. The changes in TG levels observed in rabbits receiving LPS could have similar consequences.

Experimentally induced periodontitis has also been reported to cause multiple alterations in the liver of rats compatible with NASH features in humans [62,76,77,78]. Here, the histological evaluation of livers from HFD-fed animals found a higher score of acinar inflammation in rabbits receiving LPS injections, despite showing similar NASH grades. Moreover, plasma GGT activity and uric acid levels suggested an increase in hepatic damage. However, in the present study, experimental periodontitis has no consequences on liver health markers or histological features for rabbits fed SDs. Similarly, in other studies where NAFLD was induced by an experimental diet, periodontitis induction had no effect [54,78] or only a very low effect [63] on liver alterations or enzymatic markers of hepatic injury in animals receiving a SD. In contrast to the observed effect in the aorta, these results suggest that, in this model, periodontal condition could promote diet-induced NAFLD progression towards NASH, but the results did not support that it may be an independent predictor for the development of NAFLD, as showed by previous studies conducted in animals [62,77,78].

## 5. Limitations and Perspective

Finally, in relation to the possible limitations of this study, although there are some differences between rabbits and humans in terms of lipoprotein metabolism [35], these differences do not seem relevant for lesions found in rabbits and humans. However, most commercially available rabbits are not inbred and can present important inter-individual variations, differently responding to the high–cholesterol diet. This could explain the absence of some statistically significant effect on some plasma lipids. Notwithstanding, rabbits were provided by the same breeders and had the same age and sex, which could minimize the variations. Regarding other differences with humans, rabbits can suffer from extraordinarily “high” hypercholesterolemia and show massive lipid accumulation in many organs when they are feeding on diets containing more than 1% cholesterol for a long period (more than a month) [35]. Future studies are needed to carry out closer monitoring of plasma lipid levels throughout the experimental period and even an analysis of the composition of circulating lipoproteins, all without compromising ethical aspects such as excessive invasiveness in experimental animals.

Concerning limitations regarding aortic lesions, maybe the effect of the diet in this tissue and the presence of foamy cells in gums are not representatives of the respective diseases in humans. Moreover, the atherosclerotic lesions in rabbits fed cholesterol diets consist almost completely of foam cells, which are seldom seen in humans [35]. Nevertheless, as in the present study, one of the major lesions are usually fatty streaks as seen in human atherosclerosis [35]. On the other hand, histopathological findings in the liver are compatible with human NAFLD and previous studies have revealed molecular mechanisms similar to those involved in the pathogenesis of human NASH in rabbits fed on a high-fat diet supplemented with 20% corn oil and 1.25% (*w*/*w*) cholesterol for 8 weeks [36]. Some histopathological aspects found in gums of other LPS-induced models of periodontitis and to that observed in established periodontitis in humans are less evident in this model [34], but for this reason, our conclusions have been focused on ABL, a major feature of human periodontitis.

All the findings found in the present study, as well as the doubts and limitations that have arisen, ensure future studies in which the relationship between diet and different systemic diseases prevalent in humans can be explored, but in an experimental model that has been shown to be useful. The use of other techniques to generate periodontitis such as gum ligation or infection with different bacterial strains or the application of bacterial products may be tested alone or in combination to deepen and improve the model, bringing it closer to the situation observed in humans.

## 6. Conclusions

Overall, the results of the present study support that certain dietary habits considered as crucial risk factors for the onset and development of dyslipidemia, CVDs, or NAFLD can exacerbate some aspects of periodontitis, but the presence of certain oral microorganisms in subgingival plaque is an indispensable factor. Moreover, this study also suggests that bad periodontal health can contribute to dyslipidemia and promote NAFLD progression, triggered by “unhealthy” feeding, but it would not be an independent predictor for the development of these conditions.

## Figures and Tables

**Figure 1 nutrients-12-01405-f001:**
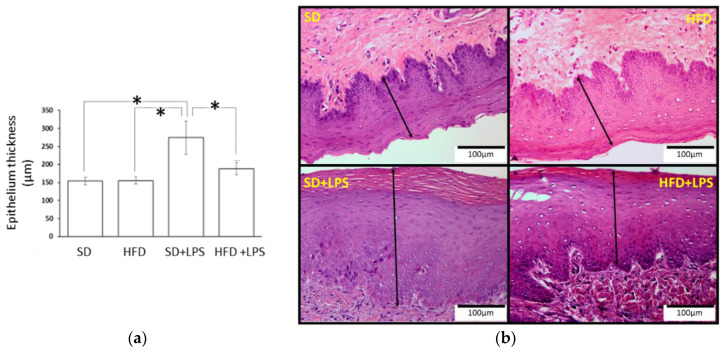
Gingival epithelium thickness. (**a**) Epithelium thickness measurements. Columns and error bars represent mean ± standard error of the mean, respectively. Asterisks (*****) represent statistically significant differences (*p* < 0.05) in thickness between the groups. (**b**) Histological section images. Dimension lines represent the thickness of the epithelium. Abbreviations: SD: Standard diet-fed rabbits non-treated with bacterial lipopolysaccharide; HFD: high fat diet-fed rabbits non-treated with bacterial lipopolysaccharide; HPF: high-power field; SD + LPS: Standard diet-fed rabbits treated with bacterial lipopolysaccharide; HFD + LPS: High fat diet-fed rabbits treated with bacterial lipopolysaccharide.

**Figure 2 nutrients-12-01405-f002:**
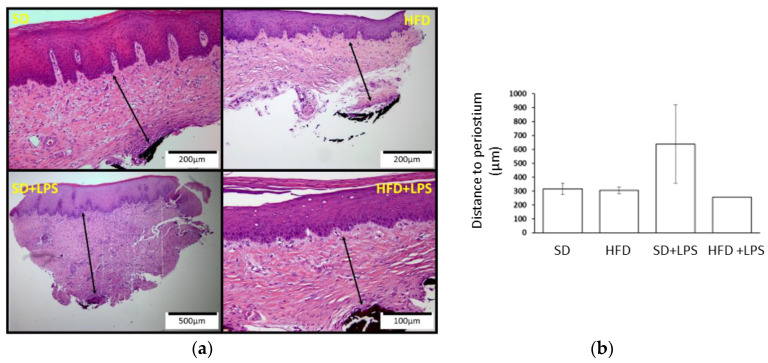
Distance from the gingival epithelium to the periosteum. (**a**) Histological section images. Dimension lines represent the distance between the gingival epithelium and the periosteum. (**b**) Distance from the gingival epithelium to the periosteum measurements. Columns and error bars represent mean ± standard error of the mean, respectively. Abbreviations: SD: Standard diet-fed rabbits non-treated with bacterial lipopolysaccharide; HFD: high fat diet-fed rabbits non-treated with bacterial lipopolysaccharide; HPF: high power field; SD + LPS: Standard diet-fed rabbits treated with bacterial lipopolysaccharide; HFD + LPS: High fat diet-fed rabbits treated with bacterial lipopolysaccharide.

**Figure 3 nutrients-12-01405-f003:**
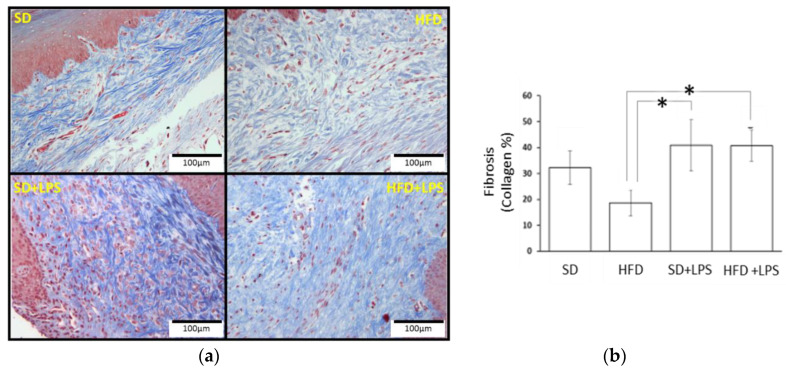
Gingival fibrosis. (**a**) Histological section images. Fibrosis of each gum sample was assessed in Masson’s Trichrome stained sections calculating the area and percentage of collagen located in the chorion of the mucosa. (**b**) Fibrosis values. Columns and error bars represent mean ± standard error of the mean, respectively. Asterisks (*****) represent statistically significant differences (*p* < 0.05) in the line between the groups. Abbreviations: SD: Standard diet-fed rabbits non-treated with bacterial lipopolysaccharide; HFD: high fat diet-fed rabbits non-treated with bacterial lipopolysaccharide; HPF: high power field; SD + LPS: Standard diet-fed rabbits treated with bacterial lipopolysaccharide; HFD + LPS: High fat diet-fed rabbits treated with bacterial lipopolysaccharide.

**Figure 4 nutrients-12-01405-f004:**
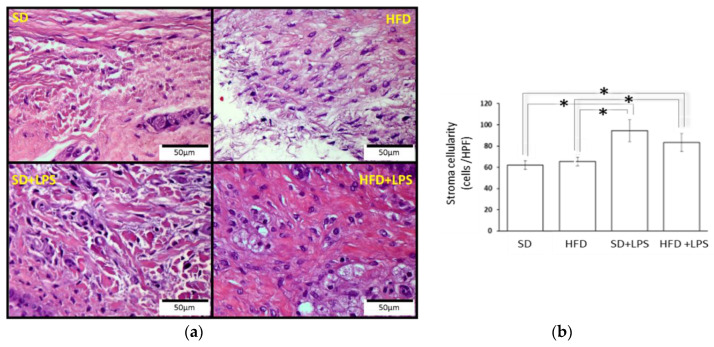
Gingival stroma cellularity. (**a**) Histological section images. (**b**) Cell amount. Columns and error bars represent mean ± standard error of the mean. Asterisks (*****) represent statistically significant differences (*p* < 0.05) in the line between the groups. Abbreviations: SD: Standard diet-fed rabbits non-treated with bacterial lipopolysaccharide; HFD: high fat diet-fed rabbits non-treated with bacterial lipopolysaccharide; HPF: high power field; SD + LPS: Standard diet-fed rabbits treated with bacterial lipopolysaccharide; HFD + LPS: High fat diet-fed rabbits treated with bacterial lipopolysaccharide.

**Figure 5 nutrients-12-01405-f005:**
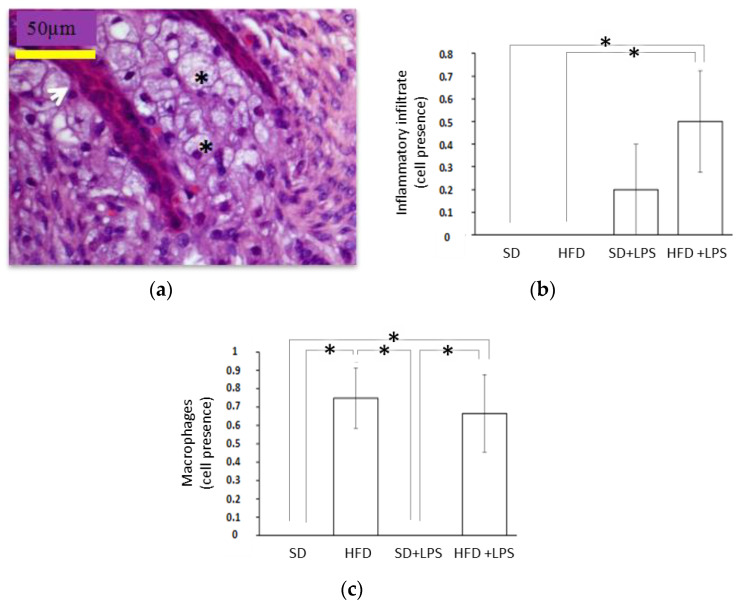
Histological features in gingival tissue II. (**a**) Chorion deposits of foamy macrophages. The black arrow is on the baseline epithelium of the gingiva. Hematoxylin and eosin 40X; (**b**) Macrophages count in the gingiva of the rabbits; (**c**) Gingival inflammatory infiltrate; Abbreviations: SD: Standard diet-fed rabbits non-treated with bacterial lipopolysaccharide; HFD: high fat diet-fed rabbits non-treated with bacterial lipopolysaccharide; HPF: high-power field; SD + LPS: Standard diet-fed rabbits treated with bacterial lipopolysaccharide; HFD+LPS: High fat diet-fed rabbits treated with bacterial lipopolysaccharide. Columns and error bars represent mean ± standard error of the mean, respectively. Asterisks (*****) represent statistically significant differences (*p* < 0.05) in the line between the groups.

**Figure 6 nutrients-12-01405-f006:**
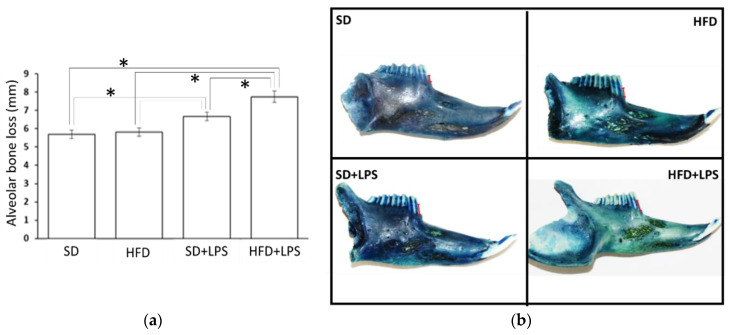
Alveolar bone losses in rabbits. (**a**) Alveolar bone loss in the experimental groups. Columns and error bars represent mean ± standard error of the mean, respectively. Asterisks (*****) represent statistically significant differences (*p* < 0.05) in the line between the groups. (**b**) Images of the jaws stained with methylene blue show how bone resorption has been measured from the top of the cementoenamel junction of the tooth (upper part of the red mark) to the alveolar bone crest (lower part of the red mark). Abbreviations: SD: Standard diet-fed rabbits non-treated with bacterial lipopolysaccharide; HFD: high fat diet-fed rabbits non-treated with bacterial lipopolysaccharide; SD + LPS: Standard diet-fed rabbits treated with bacterial lipopolysaccharide; HFD + LPS: High fat diet-fed rabbits treated with bacterial lipopolysaccharide.

**Table 1 nutrients-12-01405-t001:** Blood biochemistry.

Marker	SD	HFD	SD + LPS	HFD + LPS
Glucose (mg/dL)	158.3 ± 21.0 ^a^	348.0 ± 28.3 ^b^	129.1 ± 20.5 ^a^	424.9 ± 41.3 ^b^
Triglycerides (mg/dL)	89.9 ± 15.7 ^a^	386.2 ± 51.0 ^b^	55.0 ± 9.9 ^a^	845.0 ± 254.2 ^c^
Phospholipids (mg/dL)	63.1 ± 4.6 ^a^	586.2 ± 82.6 ^b^	57.5 ± 3.6 ^a^	915.9 ± 254.2 ^c^
LDL-cholesterol (mg/dL)	1.3 ± 0.3 ^a^	113.7 ± 5.2 ^b^	2.7 ± 0.6 ^a^	139.7 ± 14.3 ^b^
HDL-cholesterol (mg/dL)	16.6 ± 1.3 ^a^	138.7 ± 5.2 ^b^	14.5 ± 2.2 ^a^	187.4 ± 34.5 ^b^
Total cholesterol (mg/dL)	89.9 ± 15.7 ^a^	386.2 ± 51.0 ^b^	55.0 ± 9.9 ^a^	845.0 ± 254.2 ^b^
LDH (U/L)	334.3 ± 36.0	303.0 ± 61.6	379.1 ± 100.6	208.9 ± 59.6
GGT (U/L)	9.0 ± 0.2 ^a^	17.1 ± 6.0 ^a^	7.8 ± 1.2 ^a^	57.7 ± 17.6 ^b^
ALT (U/L)	28.5 ± 8.2	56.5 ± 21.7	28.8 ± 4.9	25.8 ± 10.2
AST (U/L)	42.4 ± 8.3	53.7 ± 11.4	49.3 ± 15.7	66.1 ± 16.9
Total bilirubin (mg/dL)	2.8 ± 1.1 ^a,b^	5.5 ± 1.1 ^b^	1.6 ± 0.2 ^a^	10.6 ± 2.4 ^c^
Direct bilirubin (mg/dL)	0.6 ± 0.0 ^a^	3.2 ± 0.6 ^b^	0.4 ± 0.0 ^a^	3.7 ± 0.8 ^b^
Creatinine (mg/dL)	1.2 ± 0.1 ^a^	2.7 ± 0.2 ^b^	1.5 ± 0.1 ^a^	2.4 ± 0.7 ^a,b^
Uric acid (mg/dL)	1.3 ± 0.7 ^a^	34.0 ± 6.5 ^b^	0.7 ± 0.2 ^a^	65.4 ± 15.4 ^c^
Urea (mg/dL)	15.9 ± 1.1 ^a^	11.0 ± 8.2 ^a,b^	17.8 ± 2.5 ^a,b^	22.8 ± 2.4 ^b^

Abbreviations: AU, Arbitrary units; SD, Standard diet-fed rabbits non-treated with bacterial lipopolysaccharide; HFD, high fat diet-fed rabbits non-treated with bacterial lipopolysaccharide; HDL, high-density lipoprotein; SD + LPS, Standard diet-fed rabbits treated with bacterial lipopolysaccharide; HFD + LPS, High fat diet-fed rabbits treated with bacterial lipopolysaccharide; GGT, gamma-glutamyl transferase; AST, aspartate transaminase; ALT: alanine transaminase; CK, creatine kinase; LDH, lactate dehydrogenase; LDL: low-density lipoprotein. Results are expressed as mean ± standard error of the mean. Different superscript letters indicate statistically significant differences between experimental groups (*p* < 0.05).

**Table 2 nutrients-12-01405-t002:** Histopathological analysis of aorta and liver.

	SD	HFD	SD + LPS	HFD + LPS
**Aortic lesions**				
Aortic arch (AU)	0.00 ± 0.00 ^a^	2.38 ± 0.32 ^c^	0.00 ± 0.00 ^a^	1.50 ± 0.22 ^b^
Thoracic aorta (AU)	0.00 ± 0.00 ^a^	1.00 ± 0.19 ^c^	0.00 ± 0.00 ^a^	0.33 ± 0.21 ^b^
Abdominal aorta (AU)	0.00 ± 0.00 ^a^	0.25 ± 0.16 ^b^	0.00 ± 0.00 ^a^	0.43 ± 0.22 ^b^
**Hepatic Lesions**				
Steatosis (hepatocytes %)	0.5 ± 0.5 ^a^	65.25 ± 7.09 ^b^	2.25 ± 1.48 ^a^	73.57 ± 19.11 ^b^
Lobular inflammation (inflammatory foci/20 HPF)	0.38 ± 0.26 ^a^	1.5 ± 0.54 ^b^	0.63 ± 0.63 ^a^	3 ± 0.57 ^c^
Ballooning degeneration (AU)	0.75 ± 0.31 ^a,b^	1.63 ± 0.18 ^b^	0 ± 0 ^a^	1.4 ± 0.29 ^b^
NASH score (AU)	1 ± 0.42 ^a^	5.125 ± 0.52 ^b^	0.63 ± 0.38 ^a^	5.86 ± 0.67 ^b^

Abbreviations: AU, Arbitrary units; SD, Standard diet-fed rabbits non-treated with bacterial lipopolysaccharide; HFD, high fat diet-fed rabbits non-treated with bacterial lipopolysaccharide; SD + LPS, Standard diet-fed rabbits treated with bacterial lipopolysaccharide; HFD + LPS, High fat diet-fed rabbits treated with bacterial lipopolysaccharide; HPF, high-power field. Results are expressed as mean ± standard error of the mean. Different superscript letters indicate statistically significant differences between experimental groups (*p* < 0.05).

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
