# Peer review of "A Diet Rich in Saturated Fat and Cholesterol Aggravates the Effect of Bacterial Lipopolysaccharide on Alveolar Bone Loss in a Rabbit Model of Periodontal Disease"

_nutrients, 2020, doi:10.3390/nu12051405_

Round 1

Reviewer 1 Report

These authors reported that HFD enhanced LPS-induced periodontitis in a rabbit model. Although the study was performed based on the notation that there is a relationship between periodontitis and many metabolic diseases, it is not necessary to think that all these relations are causal. The major concerns they should consider are as follows.

  1. I am not sure at which time point you measure the blood parameters (Table 1). Normally, you should measure weekly or biweekly. Why LPS increases plasma lipids?
  2. It is not clear how you measure the lesions of aortic atherosclerosis. 
  3.  I cannot see any inflammation in the alveolar bone loss with my eyes. Quantification methods are questionable. 

Author Response

These authors reported that HFD enhanced LPS-induced periodontitis in a rabbit model. Although the study was performed based on the notation that there is a relationship between periodontitis and many metabolic diseases, it is not necessary to think that all these relations are causal. The major concerns they should consider are as follows.

Reviewer: I am not sure at which time point you measure the blood parameters (Table 1). Normally, you should measure weekly or biweekly. Why LPS increases plasma lipids?

Authors: The experimental model based on the administration of saturated fat and cholesterol to induce atherosclerosis and blood lipids alterations in rabbits is very well known. So, the course of the disease through time is well stablished. For the present study we decided to test this model, which has also shown in the past to be effective for the induction of NASH, for the study of periodontal features. For this reason, the study was designed in terms of endpoint, not control over time throughout the study. We considered that the obtention of additional measurement points was not going to be relevant for the purpose of the study. In addition, it was avoided to generate additional stress to the animals, which already had to be manipulated for other experimental procedures.

Concerning the effect of LPS on blood lipids, as in other models, it is expected that LPS administration triggers a release of multiple inflammatory mediators that directly or indirectly would modulate lipid metabolism in different tissues (for example, Feingold et al., 1992; Spitzer & Spitzer, 1983; Guo et al., 2015; Kallio et al., 2008) with important consequences in plama lipid levels. Notwithstanding, in our study, a significant increase in plasma levels was only found for triglycerides and phospholipids in animals received LPS and only in those maintained on a high-fat diet.

Reviewer: It is not clear how you measure the lesions of aortic atherosclerosis. 

Authors: Following reviewer recommentarions authors have included additional information in relation to the measurement of the aortic lessions: “Paraffin-embedded tissue sections of aorta and liver were stained and evaluated as in previous studies (Quiles et al., 2002). The presence of the fatty streak was evaluated in aorta samples and lesions were scored on a 4-point–intensity semiquantitative scale (1, absence of damage; 2, mild damage; 3, moderate damage; and 4, intense damage).” A reference explaining the methodology more extensively has been added.

Reviewer: I cannot see any inflammation in the alveolar bone loss with my eyes. Quantification methods are questionable.

Authors: The measurements in alveolar bone are not related to inflammation but with bone loss. Authors have included new pictures representative for all experimental groups in the new Figure 6 in order to clarify the situation.

References:

Quiles JL, Mesa MD, Ramírez-Tortosa CL, Aguilera CM, Battino M, Gil A, Ramírez-Tortosa MC. Curcuma longa extract supplementation reduces oxidative stress and attenuates aortic fatty streak development in rabbits. Arterioscler Thromb Vasc Biol. 2002 Jul 1;22(7):1225-31

Feingold KR, Staprans I, Memon RA, Moser AH, Shigenaga JK, Doerrler W, Dinarello CA, Grunfeld C. Endotoxin rapidly induces changes in lipid metabolism that produce hypertriglyceridemia: low doses stimulate hepatic triglyceride production while high doses inhibit clearance. J Lipid Res. 1992 Dec;33(12):1765-76

Judy A. Spitzer; John J. Spitzer. Effect of LPS on Carbohydrate and Lipid Metabolism in: Nowotny (ed.), Beneficial Effects of Endotoxins, 1983, pp 57-74 © Plenum Press, New York Beneficial Effects of Endotoxin

Jun Guo, Zhiqing Liu, Hailin Sun, Yanping Huang, Elke Albrecht, Ruqian Zhao & Xiaojing Yang. Lipopolysaccharide challenge significantly influences lipid metabolism and proteome of white adipose tissue in growing pigs. Lipids in Health and Disease 14, 68 (2015)

Kallio KAE, Buhlin K, Jauhiainen M. Lipopolysaccharide associates with pro-atherogenic lipoproteins in periodontitis patients, Innate Immunity 2008, 14, 4, 247-253

Reviewer 2 Report

This manuscript established a LPS-induced periodontitis model in rabbits for investigating the role of diets in saturated fat and cholesterol on periodontal disease as well as the influence on hyperlipidemia, atherosclerosis and non-alcoholic fatty liver disease. Exacerbated alveolar bone loss, a higher score of acinar inflammation at liver, higher blood levels of triglycerides and phospholipids were found in HFD-fed rabbits receiving LPS intragingival injections. This report upholds that dietary habits plays a role in exacerbating some aspects of periodontitis and a bad periodontal health might contribute to dyslipidemia and promote NAFLD progression. The experiment design is logic and well-thoughted. The data was presented clear to support the conclusion. One minor concern is that it would be ideal if the resolution of the picture can be improved.

Author Response

Reviewer: This manuscript established a LPS-induced periodontitis model in rabbits for investigating the role of diets in saturated fat and cholesterol on periodontal disease as well as the influence on hyperlipidemia, atherosclerosis and non-alcoholic fatty liver disease. Exacerbated alveolar bone loss, a higher score of acinar inflammation at liver, higher blood levels of triglycerides and phospholipids were found in HFD-fed rabbits receiving LPS intragingival injections. This report upholds that dietary habits plays a role in exacerbating some aspects of periodontitis and a bad periodontal health might contribute to dyslipidemia and promote NAFLD progression. The experiment design is logic and well-thoughted. The data was presented clear to support the conclusion. One minor concern is that it would be ideal if the resolution of the picture can be improved.

Authors: Authors acknowledge to the reviewer the appreciations about the manuscript. Concerning the resolution of the pictures it may be depending on the fact that the version to review does not have the final quality. In any case, the authors will be pending requests from the editors in case the manuscript is finally accepted, and images of greater clarity are required.

Reviewer 3 Report

  1. The authors experimentally showed that LPS induced severe alveolar bone loss in the first premolar of HFD-fed rabbits. It is an interesting research, but there is a problem in how to show the results: the HFD induced NASH, and LPS further enhanced it. There has been considerable research into the NASH mechanism. However, the authors have not been able to analyze the mechanism of alveolar bone loss. The authors only infer the relationship with alveolar bone loss from the evaluation of l NASH indexes and blood lipid levels only the relationship
  2. In the figures,” a” an “ b” were used, but it was unclear what they indicate. The symbols “(a)” and “(b)” used in the figure may be confused with the others, so it is better to use other symbols. In addition, please address what the columns and bars shown in the figure do. The authors used abbreviation of SD for standard diet and standard deviation. The vertical axis label in each figure was too small to read.
  3. What did the numbers in Tables 1 and 2 indicate? Mean and standard deviation? Please write in each legend so that the reader can see it immediately
  4. Mark the groups with significant differences in Figure 3.

Author Response

Reviewer: The authors experimentally showed that LPS induced severe alveolar bone loss in the first premolar of HFD-fed rabbits. It is an interesting research, but there is a problem in how to show the results: the HFD induced NASH, and LPS further enhanced it. There has been considerable research into the NASH mechanism. However, the authors have not been able to analyze the mechanism of alveolar bone loss. The authors only infer the relationship with alveolar bone loss from the evaluation of l NASH indexes and blood lipid levels only the relationship

Authors: It was performed a histopathological evaluation of all, aorta, liver and gingiva. NASH is a quantitative index that combine several histological parameters, not is based on one only index (Yeh & Brunt, 1999). Periodontal health evaluation was more extensively evaluated since alveolar bone loss was also measured. The effects of the used diet on liver an cardiovascular health as well known (Ramirez-Tortosa et al., 2009, 1999) and their consequences have been related to hyperlipidemia observed in these animals (Fan, et al. 2015; Ogawa, et al. 2010; Sanches, et al., 2020). Mechanisms under alveolar bone loss have not been analyzed because this was a preliminary study, but the experimental design allows to attribute to the high-fat diet consumption the increase in LPS-induced alveolar bone loss as well to related the higher values of acinar inflammation found in animals receiving a high fat diet to LPS administration, which are the main conclusions of the study.

Reviewer: In the figures,” a” an “b” were used, but it was unclear what they indicate. The symbols “(a)” and “(b)” used in the figure may be confused with the others, so it is better to use other symbols. In addition, please address what the columns and bars shown in the figure do. The authors used abbreviation of SD for standard diet and standard deviation.The vertical axis label in each figure was too small to read.

Authors: No abbreviation has been used for SD (standard deviation) in our study. In fact, the term standard deviation does not appear in the manuscript. On the other hand, the use of “a” and “b” to indicate significant differences in figures has been avoided, axis label sizes in each figure have been increased and the meaning of columns and bars has been added to the legend according to reviewer’s suggestions.

Reviewer: What did the numbers in Tables 1 and 2 indicate? Mean and standard deviation? Please write in each legend so that the reader can see it immediately

Authors: Following Reviewer’s suggestion, the meaning of the numbers have been added to the legend.

Reviewer: Mark the groups with significant differences in Figure 3.

Authors: Done

References:

Fan, J.; Kitajima, S.; Watanabe, T.; Xu, J.; Zhang, J.; Liu, E.; Chen, Y.E. Rabbit models for the study of human atherosclerosis: from pathophysiological mechanisms to translational medicine. Pharmacol Ther 2015, 0, 104–119.

Ogawa, T.; Fujii, H.; Yoshizato, K.; Kawada, N. A human-type nonalcoholic steatohepatitis model with advanced fibrosis in rabbits. Am. J. Pathol. 2010, 177, 153–165.

Sanches, S.C.L.; Ramalho, L.N.Z.; Augusto, M.J.; da Silva, D.M.; Ramalho, F.S. Nonalcoholic Steatohepatitis: A Search for Factual Animal Models. Available online: https://www.hindawi.com/journals/bmri/2015/574832/ (accessed on Jan 6, 2020).

Ramirez-Tortosa, M.C.; Ramirez-Tortosa, C.L.; Mesa, M.D.; Granados, S.; Gil, Á.; Quiles, J.L. Curcumin ameliorates rabbits’s steatohepatitis via respiratory chain, oxidative stress, and TNF-α. Free Radical Biology and Medicine 2009, 47, 924–931.

Ramírez-Tortosa, M.C.; Mesa, M.D.; Aguilera, M.C.; Quiles, J.L.; Baró, L.; Ramirez-Tortosa, C.L.; Martinez-Victoria, E.; Gil, A. Oral administration of a turmeric extract inhibits LDL oxidation and has hypocholesterolemic effects in rabbits with experimental atherosclerosis. Atherosclerosis 1999, 147, 371–378.

Yeh, M.M.; Brunt, E.M. Pathology of nonalcoholic fatty liver disease. American journal of clinical pathology 2007, 128, 837–847.

Reviewer 4 Report

Comments on Varela-López et al.

This is an interesting study. The results are well interpreted. However, the discussion section has been descriptive; it should be concise. Furthermore, it is hard to find out the direct link between periodontitis and other pathological conditions (such as dyslipidemia,  CVDs  or  NAFLD). Below there are some points that might be useful.

  1. The title is not reflecting the abstract. It needs to be revised.
  2. In the introduction section, there is a lack of information regarding the molecular mechanisms of periodontitis.
  3. Quantification inflammatory mediators (such as cytokines and chemokines) in LPS-induced periodontitis might be an important addition to this study.
  4. In Figures 1 & 2, representative images for all the groups of respective figures should be added.
  5. Minor spell check is required. For example, in line 221, “Macrophagues” should be replaced with “Macrophages”.

Author Response

Reviewer: This is an interesting study. The results are well interpreted. However, the discussion section has been descriptive; it should be concise. Furthermore, it is hard to find out the direct link between periodontitis and other pathological conditions (such as dyslipidemia,  CVDs  or  NAFLD). Below there are some points that might be useful.

Reviewer: The title is not reflecting the abstract. It needs to be revised.

Authors: The abstract has been reviewed and modified to improve its concordance with the title.

Reviewer: In the introduction section, there is a lack of information regarding the molecular mechanisms of periodontitis.

Authors: Following Reviewer’s suggestion, some information has been included now as follows (Page 2, lines 55-63):

“Such inflammatory response results from multiple proinflammatory cytokines and chemokines produced by periodontal tissue cells in response to subgingival plaque bacteria components [2]. Among other, accumulated inflammatory mediators would recruit polymorphonuclear leukocytes [2] that produce reactive oxygen species (ROS) through NADPH oxidase [3]. In addition, proinflammatory cytokines also would lead to an increased mitochondrial production of ROS [4]. Persistent production of ROS causes induction of oxidative stress and may trigger cell death, inflammatory responses, and perturbation of tissue homeostasis [5-7]. In fact, there is accumulating evidence indicating that oxidative stress plays a central role in the periodontal tissue and alveolar bone destruction [8-10].”

Reviewer: Quantification inflammatory mediators (such as cytokines and chemokines) in LPS-induced periodontitis might be an important addition to this study

Authors: We agree with the possible relevance of different cytokines for a better understanding of the interactions of these diseases and some of their risk factors observed in this study. However, this was a preliminary study to explore the possible existence of interactions in a new model of endotoxin induced periodontal disease in rabbit given the utility for cardiovascular and hepatic diseases research.

Reviewer: In Figures 1 & 2, representative images for all the groups of respective figures should be added.

Authors: Following Reviewer’s suggestion figure 1 has been modified and there are now four new figures. However, old figure 2 has been left as it was since differences in the histology pictures are very narrow, with only the presence or absence of macrophages and inflammatory infiltrate so we decided to left only a sample of these cells in a picture and keeping the bar charts with the quantifications.

Reviewer: Minor spell check is required. For example, in line 221, “Macrophagues” should be replaced with “Macrophages”.

Authors: Done.

Round 2

Reviewer 1 Report

Plasma lipids should be shown weekly and lipoprotein profiles should be analyzed also. There are multiple abnormalities in HDF groups addition to lipids. Aortic lesions and fatty livers should be shown histologically. The current methods for lesion analysis are not acceptable.

Author Response

Reviewer: Plasma lipids should be shown weekly and lipoprotein profiles should be analyzed also.

Authors: As commented in the first round of revision, the objective of the study was not a deep analysis of blood lipids into the rabbits, that’s why authors only measured lipids as a control of the situation at the end point of the study. In addition to that, the ethical committee approved only to take samples at the end of the study in order to avoid unnecessary pain and suffering to the animals, based on the objectives of the experiments. Concerning the question about the investigation of the lipoprotein profiles. This is completely out of the scope of the study. Furthermore, the authors are surprised by the fact that the reviewer adds new requirements not previously performed.

Reviewer: There are multiple abnormalities in HDF groups addition to lipids.

Authors: The authors do not understand the reviewer's unspecific comment at all. We do not understand what the reviewer refers to with the "multiple abnormalities".

Reviewer: Aortic lesions and fatty livers should be shown histologically.

Authors: As has been mentioned repeatedly, the main objective of the work is to study what happens in the gingival tissue of the rabbits. The appearance of lesions in the aorta and liver is nothing new, they simply had to be verified to ensure that the model was reproducing as in previous studies. For this reason, the pathologist made the determinations directly on the microscope, without taking photographs.

In the event that the reviewer requires the submission of photographs of the liver and aorta of the rabbits as an essential condition for acceptance of the manuscript, we would have to look for the samples in the laboratory and take these photographs. This will need an approximate period of 2 weeks, given that due to the current situation of COVID-19, research laboratories in Spain are closed and their access is prohibited, and it is expected that access to researchers will be authorized, probably in about a week.

Reviewer: The current methods for lesion analysis are not acceptable.

Authors: All methods used for lesion analyses have been previously validated and are based on literature. For aortic lesions, the methods previously published by our group has been used. As for the pathological features of liver for which the method of Yeh and Brunt has been used. Concerning gingiva, we have also used methods previously published. We have added now a reference for this method. Finally, the measurement of alveolar bone loss has been also measured based on previously published and validated methods. We have added a new reference to this method.

Reviewer 3 Report

None

Author Response

This Reviewer has no questions.